# Defining the Practice of Dietitians in Malaysia Using the Nutrition Care Process in Patients with Type 2 Diabetes Mellitus

Jamilah Abd Jamil [1], Esther F. Myers [2] and Winnie Chee Siew Swee [1,*]

1 Division of Nutrition and Dietetics, School of Health Sciences, International Medical University, Kuala Lumpur 57000, Malaysia
2 EF Myers Consulting, Inc., 600 N Oak Street, Trenton, IL 62293, USA
* Correspondence: winnie_chee@imu.edu.my; Tel.: +60-(03)-273-173-05

**Abstract:** (1) Background: The quality of dietetic care is crucial to improve patient outcomes. The aim of this study was to measure the current practices regarding the provision of nutrition care in type 2 diabetes mellitus (T2DM) among dietitians in Malaysia. (2) Methods: A 49-item online survey was distributed via Malaysian Dietitians' Association and Ministry of Health Malaysia social media platforms. Self-reported dietetic practices in the management of T2DM were evaluated against practice guidelines. (3) Results: A total of 173 dietitians completed the survey, with a response rate of 62%. Three-quarters of dietitians in the public sector consulted patients within two to three weeks or more than four weeks versus less than two weeks (86.3%) among private dietitians (*p*-value < 0.001). More than 95% of private dietitians spent 31–60 min or more than 60 min on new cases versus 71% among public dietitians (*p*-value < 0.001). Group counseling was only practiced among public dietitians (36.0%). Different practice settings led to the current findings. A limited use of behavioral counseling theories and strategies was observed among dietitians from both sectors. Limited adherence to nutrition-related recommendations (62.8%) and lack of readiness for diet/lifestyle changes (45.5%) were the key challenges faced by dietitians when managing T2DM. (4) Conclusions: The survey indicates that there is a need for the development of comprehensive training to increase the utilization of behavioral counselling. Practice setting is an element to consider when designing training.

**Keywords:** Medical Nutrition Therapy; Nutrition Care Process; dietetic practice; type 2 diabetes mellitus; dietitian

## 1. Introduction

Dietary management is an integral component of type 2 diabetes mellitus (T2DM) management alongside with medication and lifestyle modifications. With the rise in diabetes prevalence around the globe, dietitians must play their role to empower people with diabetes to make diet-related modifications to delay diabetes complications such as cardiovascular diseases, retinopathy, diabetic foot ulcer and chronic kidney disease [1]. In order to provide quality care to people with diabetes to facilitate dietary behavior changes, dietitians deliver Medical Nutrition Therapy (MNT) using a systematic approach called the Nutrition Care Process (NCP). Both MNT and the NCP have convincing evidence of effectiveness [2,3], as MNT can lead to significant reductions in clinical outcomes for blood glucose control and cardiovascular risks [4], while the NCP may lead to the resolution of nutrition problems by 50% [3].

Glycemic control in Malaysia continues to deteriorate despite initiatives by the Ministry of Health to increase awareness and expanded accessibility of glycosylated hemoglobin (A1c) testing across the country [5]. A major contributor to poor glycemic control is the lack of dietary adherence, along with a high consumption of carbohydrates, frequent meals

intake of four meals or more in a day and sedentary lifestyle [6,7]. Even though the country has developed evidence-based MNT guidelines and dietitians in Malaysia have implemented the NCP for more than a decade [8], little is known about the extent to which dietitians follow the standards of practice. This is a crucial aspect to measure because the quality, skills and attitudes of healthcare workers are central to the delivery of the quality of patient care [9].

The aim of this study was to conduct a dietetic practice audit with regards to the use of evidence-based guidelines and the implementation of the NCP, identify gaps in practice and determine the challenges faced by dietitians in Malaysia in delivering services for the management of T2DM.

## 2. Materials and Methods

### 2.1. Study Design

This cross-sectional, anonymous online survey was administered to dietitians and aimed to measure current practices in MNT and the NCP for T2DM management. The study protocol was reviewed and approved by International Medical University—Joint Committee on Research and Ethics (PHMS I-2020 (04)). All respondents reviewed an informed consent form and consented to participate.

### 2.2. Survey Development and Design

The questionnaire contained 49 questions distributed across six sections. Table 1 shows the different sections of the questionnaire and the questions entailed. Section A of the questionnaire sought to gather information about the dietitians' demographic details such as number of years of practice in the field and the highest degree completed. This section also included questions about dietitians' current dietetic consultation arrangements, such as the earliest consultation arranged for patients, as well as continuous professional development activities undertaken to keep up to date about diabetes management. Sections B, C, D and E covered aspects of practice within steps of the NCP, such as parameters commonly assessed among T2DM patients, the most likely and least likely nutrition diagnosis applied, and theories, models and strategies used to consult patients. Section F sought to understand practice challenges faced by dietitians when managing T2DM cases and the type of support needed to improve practice. The survey questions included a combination of open-ended and close-ended questions, Likert scales and multiple-choice options.

### 2.3. Validation of the Survey Instrument

The content validity of the survey instrument was determined based on the opinions of four experts in Malaysia. The online content validation form was emailed, and all experts were required to independently rate each question as follows: not relevant (1), item needs some revision (2), relevant but needs minor revision (3) or very relevant (4). Clear instructions were provided to facilitate the content validation process. The experts also provided written comments on how the questions could be improved. The content validity index (CVI) was calculated as content validity index for individual items (I-CVI) and content validity index for the scale (S-CVI). The relevance rating scale of 3 or 4 was recoded as 1 (of appropriate quality), and the relevance rating scale of 1 or 2 was recoded as 0 based on the binary scoring system [10].

The I-CVI was calculated for each of the questions by dividing the number of experts scoring 3 or 4 by the number of total experts involved. Of the 49 questions developed for the study, six questions received I-CVI scores of 0.75, whereas the rest of the questions received I-CVI scores of 1.00. The S-CVI was computed as S-CVI/Ave (scale-level content validity index based on the average method) and S-CVI/UA (scale-level content validity index based on the universal agreement method). The S-CVI/Ave was calculated from the total I-CVI score divided by the total number of questions, whereas the S-CVI/UA was calculated by adding the number of items that had 100% agreement and dividing that by the total number of questions [11]. The overall content validity index of the instrument

had high S-CVI values of 0.90 and 0.88 for the S-CVI/Ave and S-CVI/UA, respectively. Based on the experts' responses and comments, all questions were retained. However, the instrument was revised accordingly to improve clarity.

Once the questions were finalized, they were transferred to the SurveyMonkey.com (accessed on 15 April 2021) platform. Two experts tested out the platform using a mobile phone and a laptop to ensure that the layout was easy to read.

**Table 1.** Main questions included in the questionnaire.

| Sections | Main Questions/Types of Questions |
|---|---|
| Section A: Sociodemographic data and practice management (16 questions) | Gender, age, years of experience, highest degree obtained, certification of training for diabetes management, referring guidelines, number of patients seen in a typical week at clinics, arrangement for the earliest appointment, number of follow-ups arranged in one year, duration for individual and group consultations (new and follow-up cases). Multiple choice options. |
| Section B: Current dietetic practice in T2DM management related to nutrition assessment (5 questions) | Nutrition assessment parameters that dietitians typically assess: anthropometric measurement, biochemical data, nutrition-focused physical findings, and food-/nutrition-related history. Tick the boxes. |
| Section C: Current dietetic practice in T2DM management related to nutrition diagnosis (1 question) | Nutrition diagnosis label commonly used in managing T2DM. Likert scale. |
| Section D: Current dietetic practice in T2DM management related to nutrition intervention (12 questions) | Percentages of macronutrient prescriptions, topics covered during consultation, other alternative for dietary prescriptions such as intermittent fasting, behavioral counselling theories and strategies utilized. Tick the boxes and open-ended questions. |
| Section E: Current dietetic practice in T2DM management related to nutrition monitoring and evaluation (3 questions) | Nutrition parameters dietitians are likely to monitor. Likert scale. |
| Section F: Miscellaneous (12 questions) | Challenges in managing T2DM, types of support needed to improve practice. Tick the boxes and open-ended questions. |

### 2.4. Survey Implementation

The Krejcie and Morgan table [12] for finite population was used to determine the sample size. With a margin of error of 5% and a confidence interval of 95%, the sample size required was 278. A poster with a link to the web-based survey was prepared. To increase outreach to the dietitians, the survey was sent to MDA database through the MDA Facebook group, as well as Ministry of Health (MOH) Malaysia Whatsapp groups. The survey was made available from May 2021 till July 2021. To improve response rate, a reminder was sent every two weeks.

### 2.5. Data Management and Analysis

Raw data were downloaded from the SurveyMonkey site. The data were entered to the Statistical Packages for Social Sciences (SPSS)®, version 22. Descriptive variables were described as n (%), mean ± standard deviation (SD) and median ± interquartile range (IQR). The chi-square test was used to assess the differences in practices among dietitians in the public and private sectors. Open-ended responses on challenges in managing T2DM cases were analyzed with the thematic analysis approach as described by Braun and Clarke, 2006 [13]. First, the data were read carefully, and initial ideas were noted down. Second, responses with similar challenges were grouped together. Third, the potential themes were identified and reviewed until a consensus was reached among the investigators. The thematic analysis resulted in 11 key themes.

## 3. Results

### 3.1. Demographic Information and Professional Background

A total of 173 dietitians completed the survey; in total, 100 (57.8%) dietitians were from the public sector, and 73 (42.2%) dietitians were from the private sector. The proportion was comparable with the current database in which 60% of the dietitians work with the public sector and another 40% are in the private sector. The demographic characteristics of the participating dietitians are shown in Table 2. Participants had a median (IQR) of 33 (8) years of age, and the majority (89.6%) were females. Approximately two-thirds of participants had five years of experience in the field. Bachelor's degree (86.1%) was primarily the highest qualification obtained by the participants, because this is the entry level degree for dietetics workforce in Malaysia. Few (2.5%) dietitians obtained extra certification for diabetes management, namely, Certified Diabetes Educator and Postgraduate Diploma Certificate in Diabetes Management and Education.

**Table 2.** Demographic information and professional background of participants (*n* = 173).

| Variable | Public Sector *n* (%) | Private Sector *n* (%) | Total *n* (%) |
|---|---|---|---|
| **No. of respondents** | 100 (57.8) | 73 (42.2) | 173 (100.0) |
| **Gender** | | | |
| Male | 12 (12.0) | 6 (8.2) | 18 (10.4) |
| Female | 88 (88.0) | 67 (91.8) | 155 (89.6) |
| **Years of experience** | | | |
| Less than 5 years | 12 (12.0) | 41 (56.2) | 53 (30.6) |
| 5–10 years | 44 (44.0) | 18 (24.7) | 62 (35.8) |
| More than 10 years | 44 (44.0) | 14 (19.2) | 58 (33.5) |
| **Highest qualification** | | | |
| Bachelor's | 84 (84.0) | 65 (89.0) | 149 (86.1) |
| Master's | 14 (14.0) | 6 (8.2) | 20 (11.6) |
| Ph.D. | 2 (2.0) | 2 (2.7) | 4 (2.3) |
| **Additional certification in diabetes care** | | | |
| Yes | 1 (1.0) | 5 (6.8) | 6 (3.5) |
| No | 99 (99.0) | 68 (93.2) | 167 (96.5) |

### 3.2. Arrangement for Dietetic Consultation

Table 3 shows the current practices regarding arrangements for dietetic consultation arranged by dietitians. Evidence shows that diabetes MNT yields to greatest impact at initial diagnosis with three to four visits in three to six months lasting 45–90 min per session [14]. Results from the current survey showed that private dietitians were able to arrange the first consultation significantly sooner (less than two weeks) than public dietitians (two to four weeks) with a *p*-value of <0.001. Unlike in the public sector, group consultation was not a practice in the private sector (*p*-value < 0.001). Dietitians in private and public sectors managed 9 (13) and 14 (17) cases per week at 31–60 min for newly referred cases and less than 30 min for follow-up cases, respectively. Dietitians arranged one to two follow-ups per year (61.3%, private; 56.7%, public). The complexity of cases (83%), poor glycemic control (43%) and unresolved nutrition diagnosis (41%) were the main drivers influencing dietitians' decision to arrange follow-up sessions.

**Table 3.** Arrangement for dietetic consultation.

| Variable | Public (*n* = 100) *n* (%) | Private (*n* = 73) *n* (%) | *p*-Value |
|---|---|---|---|
| **Earliest appointment** | | | |
| Less than 2 weeks | 30 (30.0) | 63 (86.3) | |
| 2–4 weeks | 47 (47.0) | 2 (2.7) | * <0.001 |
| More than 4 weeks | 23 (23.0 | 8 (11.0) | |
| **No. of follow-ups** | | | |
| 1–2 follow-up | 57 (57.0) | 46 (63.0) | |
| 3–4 follow-ups | 43 (43.0) | 27 (37.0) | 0.426 |
| **Arrangement for consultation** | | | |
| Offers individual consultation | 96 (96.0) | 72 (98.6) | 0.308 |
| Offers group consultation | 36 (36.0) | 0 (0.0) | * <0.001 |
| Offers both individual and group consultation | 39 (39.0) | 0 (0.0) | * <0.001 |
| Offers interprofessional diabetes clinic | 9 (9.0) | 9 (12.3) | 0.479 |
| **Duration for individual consultation (new case)** | | | |
| Less than 30 min | 29 (29.0) | 3 (4.1) | |
| 31–60 min | 71 (71.0) | 60 (82.2) | * <0.001 |
| More than 60 min | 0 (0.0) | 10 (13.7) | |
| **Duration for individual consultation (follow-up case)** | | | |
| Less than 30 min | 91 (91.0) | 47 (64.4) | |
| 31–60 min | 8 (8.0) | 25 (34.2) | * <0.001 |
| More than 60 min | 1 (1.0) | 1 (1.4) | |

* *p*-value of <0.05 was considered significant.

### 3.3. Self-Reported Dietetic Practice in T2DM Management

Dietetic practices within the nutrition assessment steps are shown in Table 4. Measured weight and body mass index (89.6%) were the main anthropometric measurements assessed by the dietitians. A small proportion of dietitians assessed body composition (36.4%) and waist circumference (17.3%). Blood pressure (76.3%) was indicated as the most measured parameter among the nutrition-focused physical findings. All dietitians (100%) reported glycemic control as the most frequently assessed biochemical data, followed by self-monitoring blood glucose (SMBG) (94.2%) recorded by patients and the renal profile (80.3%). For food-/nutrition-related history, the three most measured parameters were dietary intake (100%), readiness to change (92.5%) and physical activity (91.9%). Knowledge of diet and diabetes was the least frequently assessed parameter by the dietitians (53.8%). No significant difference was observed in terms of variation in practices between public and private dietitians for nutrition assessment (*p*-value > 0.05).

**Table 4.** Dietetic practice within the NCP.

| Item | Public (*n* = 100)<br>*n* (%) | Private (*n* = 73)<br>*n* (%) | Total (*n* = 173)<br>*n* (%) | *p*-Value |
|---|---|---|---|---|
| **NUTRITION ASSESSMENT** | | | | |
| **Anthropometric measurement** | | | | |
| Weight | 92 (92.0) | 70 (95.6) | 162 (93.6) | 0.300 |
| Body mass index (BMI) | 87 (87.0) | 68 (93.2) | 155 (89.6) | 0.191 |
| Body composition | 36 (36.0) | 27 (37.0) | 63 (36.4) | 0.894 |
| Waist circumference | 16 (16.0) | 14 (19.2) | 30 (17.3) | 0.586 |
| **Biochemical data** | | | | |
| Glycemic control | 100 (100.0) | 73 (100.0) | 173 (100) | – |
| Self-monitoring blood glucose | 95 (95.0) | 68 (93.2) | 163 (94.2) | 0.607 |
| Renal profile | 80 (80.0) | 59 (80.8) | 139 (80.3) | 0.893 |
| Fasting blood lipids | 72 (80.0) | 45 (72.6) | 117 (67.6) | 0.254 |
| **Nutrition-focused physical findings** | | | | |
| Fat loss | 35 (35.0) | 31 (42.5) | 66 (38.2) | 0.318 |
| Blood pressure | 79 (79.0) | 53 (72.6) | 132 (76.3) | 0.328 |
| **Food-/nutrition-related history** | | | | |
| Dietary intake | 100 (100.0) | 73 (100.0) | 173 (100) | – |
| Food and nutrition knowledge | 57 (57.0) | 39 (53.4) | 96 (55.5) | 0.640 |
| Readiness to change | 89 (89.0) | 71 (97.3) | 160 (92.5) | 0.120 |
| Survival skills/nutrition self-management | 66 (66.0) | 52 (71.2) | 118 (68.2) | 0.465 |
| Living situation | 81 (81.0) | 67 (91.8) | 148 (85.5) | 0.046 |
| Physical activity | 91 (91.0) | 68 (93.2) | 159 (91.9) | 0.608 |
| Medication | 90 (90.0) | 66 (90.4) | 156 (90.2) | 0.929 |
| **NUTRITION INTERVENTION** | | | | |
| **Carbohydrate prescription** | | | | |
| Low (44% or less) | 0 (0.0) | 5 (6.8) | 5 (2.9) | |
| Moderate (45–50%) | 81 (81.0) | 60 (82.2) | 141 (81.5) | * 0.027 |
| High (>50–60%) | 17 (17.0) | 5 (6.8) | 22 (12.7) | |
| No specific prescription | 2 (2.0) | 3 (4.1) | 5 (2.9) | |
| **Protein prescription** | | | | |
| 15–20% | 81 (82.0) | 46 (63.0) | 127 (73.4) | |
| 20–25% | 10 (10.0) | 10 (13.7) | 20 (11.6) | * 0.024 |
| >25–30% | 3 (3.0) | 7 (9.6) | 10 (5.8) | |
| No specific prescription | 8 (8.0) | 8 (11.0) | 16 (9.2) | |

**Table 4.** *Cont.*

| Item | Public (*n* = 100) n (%) | Private (*n* = 73) n (%) | Total (*n* = 173) n (%) | *p*-Value |
|---|---|---|---|---|
| **Fat prescription** | | | | |
| 25–30% | 89 (89.0) | 61 (83.6) | 150 (86.7) | |
| >30–40% | 3 (3.0) | 3 (4.1) | 6 (3.5) | 0.559 |
| >40–50% | 0 (0.0) | 1 (1.4) | 1 (0.6) | |
| No specific prescription | 8 (8.0) | 8 (11.0) | 16 (9.2) | |
| **Other prescription** | | | | |
| Intermittent fasting | 10 (10.0) | 6 (8.2) | 16 (9.2) | 0.690 |
| Plant-based | 2 (2.0) | 10 (13.7) | 12 (6.9) | * 0.003 |
| Vitamin and mineral supplements | 5 (5.0) | 10 (13.7) | 15 (8.7) | * 0.045 |
| Meal replacement | 80 (80.0) | 59 (80.8) | 139 (80.3) | 0.893 |
| Total meal replacement | 8 (8.0) | 10 (13.7) | 18 (10.4) | 0.225 |
| None | 16 (16.0) | 63 (86.3) | 79 (45.7) | 0.676 |
| **Tools used to teach portion size** | | | | |
| Household measurement | 98 (98.0) | 65 (89.0) | 163 (94.2) | 0.013 |
| Hand | 62 (62.0) | 46 (63.0) | 108 (62.4) | 0.892 |
| Food model | 44 (44.0) | 29 (39.7) | 73 (42.2) | 0.574 |
| Food album | 45 (45.0) | 22 (30.1) | 67 (38.7) | * 0.047 |
| Menu plan | 43 (43.0) | 29 (39.7) | 72 (41.6) | 0.666 |
| Food label | 61 (61.0) | 37 (50.7) | 98 (56.6) | 0.229 |
| **Use of behavioral theories** | | | | |
| Health Belief Model | 10 (10.0) | 29 (39.7) | 39 (22.5) | * <0.001 |
| Trans-Theoretical Model | 9 (9.0) | 22 (30.1) | 31 (17.9) | * <0.001 |
| Cognitive-Behavioral Theory | 32 (32.0) | 33 (45.2) | 65 (37.6) | 0.007 |
| None | 58 (58.0) | 18 (24.7) | 76 (43.9) | * <0.001 |
| **Use of counseling strategies** | | | | |
| Motivational interviewing | 46 (46.0) | 52 (71.2) | 98 (56.6) | * 0.001 |
| Goal setting | 48 (48.0) | 59 (80.8) | 107 (61.8) | * <0.001 |
| Social support | 28 (28.0) | 35 (47.9) | 63 (36.4) | * 0.007 |
| Self-monitoring | 58 (58.0) | 51 (69.9) | 109 (63.0) | 0.110 |

* *p*-value of <0.05 was considered significant.

Nutrition assessment is the first step of the NCP. It is recommended that dietitians focus nutrition assessment on dietary intake, metabolic control, weight changes and physical activity to serve as the basis for the implementation of nutrition intervention once nutrition diagnosis has been identified [15]. The current findings implied that dietitians performed nutrition assessment as per recommended by available evidence.

The second step of the NCP is nutrition diagnosis. Figure 1 shows the nutrition diagnosis labels commonly identified by dietitians when managing T2DM cases. The top five frequently used labels were excessive carbohydrate intake (98%), inconsistent carbohydrate intake (94%), food- and nutrition-related knowledge deficit (85%), intakes of types of carbohydrates inconsistent with needs (74%) and excessive energy intake

(69%). These labels are among the common nutrition diagnosis labels for individuals with T2DM [16]. Out of the 17 nutrition diagnosis labels provided, unintended weight gain and intakes of types of fats inconsistent with needs (83% and 87%, respectively) were in the category of rarely used or never used. No significant differences were observed between sectors (*p*-value > 0.05).

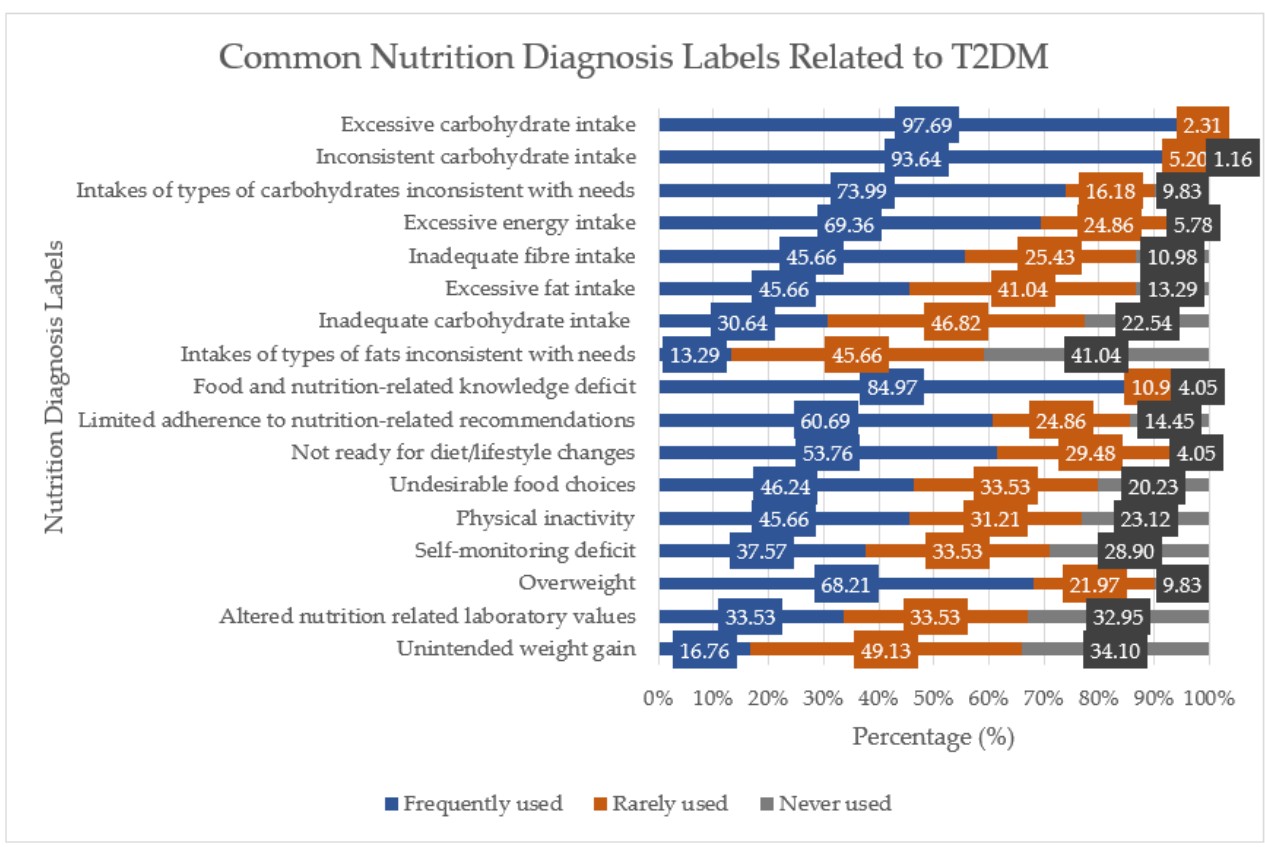

**Figure 1.** Nutrition diagnosis labels commonly used by dietitians when managing T2DM.

Dietetic practices within the nutrition intervention step are shown in Table 4. Dietitians prescribed 45–50% of carbohydrate (82%), 15–20% of protein (74%) and 25–30% of fat (87%) to T2DM patients. A small proportion (3–9%) of dietitians indicated the prescription of 'no specific range'. While there is no research to support ideal macronutrient percentages for people with diabetes, dietitians should encourage patients to consume macronutrients based on the Dietary Reference Intake [15,16]. The current prescription is within the recommendation of Malaysia's Recommended Nutrient Intake of 50–65% of carbohydrates, 15–20% of protein and 25–30% of fat of the total energy intake [17]. An alternative prescription recommended by the majority of the dietitians was meal replacement (83%). Plant-based meals (6.9%), vitamin and mineral supplements (8.7%), intermittent fasting (9.2%) and total meal replacement (10.4%) were among the least prescribed alternatives. The dietitians were asked to rank the top three nutrition education contents covered in a consultation, and they were (i) carbohydrate portion or counting (64%), (ii) consistency in carbohydrate intake (35%) and (iii) increase in fiber intake (24%). Household measurements (94.2%) and the hand method (62.4%) were tools used by dietitians to educate patients on portion size. Dietitians reported to have applied SMBG readings to complement dietary advice to patients (84%).

Dietitians are encouraged to use behavior change theories and strategies to plan effective interventions toward clients' goals, as strong evidence exists to demonstrate the effectiveness of behavior change theories and strategies to facilitate behavior change [18]. The current survey showed that more than half of the public dietitians (57.8%) indicated

that they did not use any behavioral theories as compared with private dietitians (24.7%), with a *p*-value < 0.001. For those who incorporated behavioral theories in their consultation, cognitive behavioral theory was the more popular theory (36.7%). As for the counselling strategies, a pattern was observed in which the private dietitians tended to utilize all strategies indicated in the questionnaire, namely, motivational interviewing (*p* = 0.001), goal setting (*p* < 0.001) and social support (*p* = 0.007). It is evident that skill development training is much needed to improve the use of behavior theories and strategies among dietitians in Malaysia.

To determine the effectiveness of MNT, dietitians should monitor and evaluate food intake, medication, metabolic control (glycemia, lipids and blood pressure), anthropometric measurements and physical activity [15,16]. The survey showed that more than 90% of dietitians monitored and evaluated total carbohydrate intake, consistency in carbohydrate intake, types of carbohydrate intake and hemoglobin A1c. The majority of the dietitians (81% to 88%) were also likely to measure weight change, total estimated fiber intake, self-monitoring blood glucose and physical activity.

### 3.4. Challenges and Support Needed by Dietitians

Under the open comment section, dietitians were asked to identify the key challenges they faced when managing T2DM cases. Themes that emerged were challenges related to patients or practice (Figure 2). Limited adherence to nutrition-related recommendations was the most frequently mentioned challenge (62.8%). Other challenges related to patients were the lack of readiness for diet/lifestyle changes (45.5%), food- and nutrition-related knowledge deficit (42.8%), self-monitoring deficit (43.4%), lack of supportive environment (31.0%) and undesirable food choices (22.1%). For challenges related to practice, dietitian reported to experience limited time (11.7%), limited training (11.7%), limited resources (10.3%), limited inter-professional activities (10.3%) and challenges with management and insurance coverage (5.5%). Some of the example quotes were:

- Patients' readiness for lifestyle change;
- Patient adherence to the intervention;
- Patient's understanding for diet consultation especially among elderly;
- Patient did not do SMBG;
- Lack of family or social support;
- Lack of support from doctor/specialist;
- Counselling skill is something I need to work on;
- Dietary consultation charges might not be covered under insurance claims and poor follow up because no insurance coverage.

While the dietitians in the study reported to use several evidence-based nutrition practice guidelines when managing T2DM cases such as MDA MNT Guidelines 2013 (96%), Clinical Practice Guidelines and Malaysian Endocrine and Metabolic Society (MEMS) Guidelines (79%), the American Diabetes Association (54%) and the Evidence Analysis Library (EAL) (24%), they agreed that the support they needed to improve practice was to attend more training and education (90%). Other types of support were on resources (81%), more case discussions (49%) and support from top management (39%).

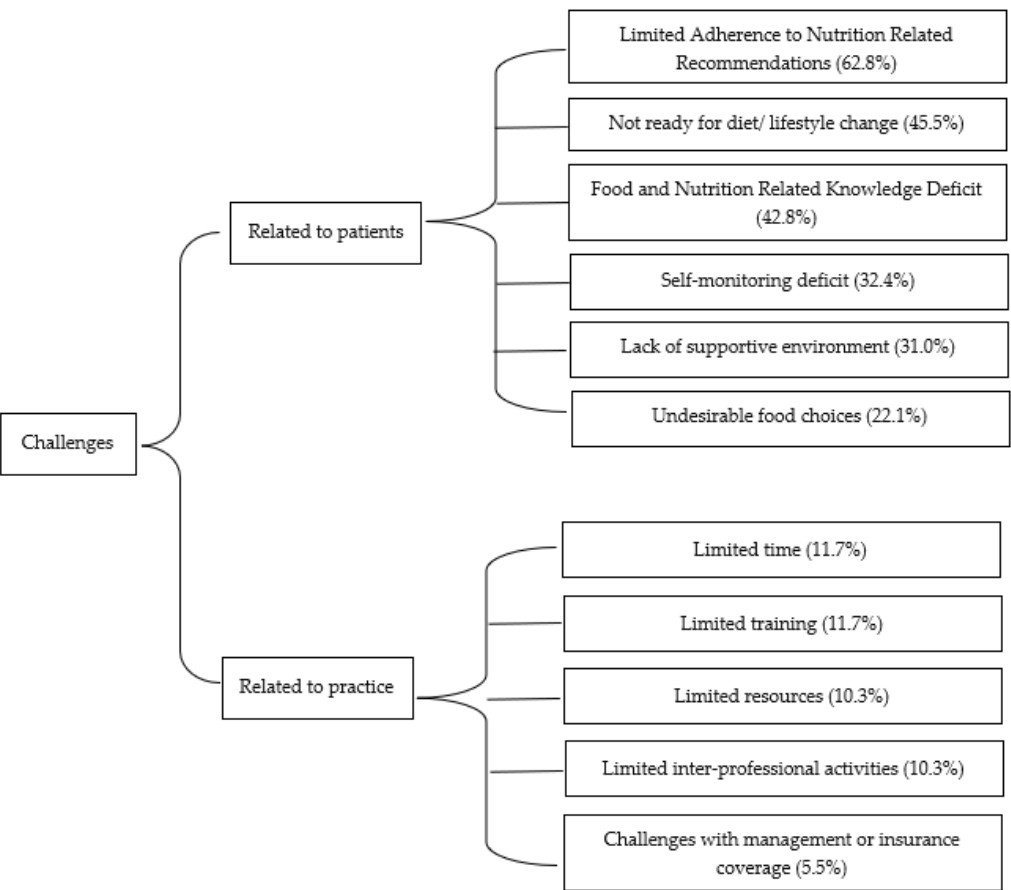

**Figure 2.** Challenges faced by dietitians when managing T2DM.

## 4. Discussion

To the best of the researchers' knowledge, this is the first study looking at practices among dietitians in Malaysia and comparing them to existing guidelines. In this study, we found that good practices were demonstrated by dietitians within nutrition assessment and nutrition monitoring, as well as evaluation steps of the NCP. The use of behavioral counseling theories and strategies was still limited among dietitians from both sectors. Due to the difference in practice settings, we identified some variations in the arrangement for dietetic consultations between the public and private sectors.

From the survey, dietitians from the public and private sectors closely followed the evidence-based guidelines when conducting nutrition assessment, and nutrition monitoring and evaluation of T2DM patients. It has been suggested that evidence-based practice guidelines and standards of practice are essential to promote the uptake of research findings into routine practice [19–21]. More than 95% of dietitians used MDA MNT Guidelines for T2DM as their main reference to manage T2DM. The reason for these good practices in nutrition assessment, and nutrition monitoring and evaluation is probably due to the availability of MDA MNT Guidelines for T2DM since 2003. Over the years, many continuing professional developments have also been conducted by various professional organizations related to diabetes management.

Four out of the five most used nutrition diagnosis labels were from the intake domain and were related to carbohydrate and energy intakes. This was expected as the staple food of Malaysians is rice, and T2DM patients in Malaysia consume 59% or 221 grams/day of carbohydrate [22], with rice contributing to 85% of dietary carbohydrates [23]. Although evidence for an optimal percentage for CHO in the food or meal plan for glycemic control is inconclusive, a consensus is emerging to classify a CHO intake of >45% as high [24].

Cardiovascular disease represents the main cause of morbidity and mortality among people with T2DM [25]. However, in this study, we found that dietitians from both sectors either 'rarely used' (45.7%) or 'never used' (41.0%) 'types of fats inconsistent with needs'. Dietitians may have prioritized the problem of carbohydrate intake among patients during the first visit and intended to address other problems such as types of fat intake in follow-up sessions. However, due to poor follow-up turn-ups as specified by dietitians in this study, this may have led to the existing results.

One interesting finding from the study was the fact that while 'food and nutrition knowledge deficit' was identified as one of the top three commonly used diagnosis labels, the assessment of patients' knowledge about diet and diabetes was the parameter being least frequently assessed by the dietitians. This is an aspect to be highlighted and must be addressed in future training.

Another main practice gap identified among both public and private sector dietitians was the limited use of behavioral counselling theories and strategies, with almost 60% dietitians reporting that they did not use any theories or strategies when consulting patients. While its utilization was generally low, we observed a pattern where dietitians from the private sector were more likely to incorporate counseling theories and strategies. This trend may be associated with the current finding indicating that dietitians in the private sector could spend more time with their patients. As described by Vasiloglou et al., counseling practices require time [26].

Diet counseling has shown to be effective in lowering HbA1c by 1–2% [27]. Strong evidence suggests that motivational interviewing is an effective counselling strategy, especially when combined with cognitive behavioral therapy (CBT) [18]. Behavioral theories and strategies are also listed as the standardized language in the electronic Nutrition Care Process Terminologies (e-NCPT) [28]. The ability to provide nutrition counselling using appropriate theories and strategies has been recognized as one of the core competencies that dietitians must have [29]. The majority of dietitians in the study reported their main challenges in managing T2DM to be patients' limited adherence to nutrition-related recommendations (62.8%) and the lack of readiness for diet/ lifestyle changes (45.5%). Interventions delivered by dietitians supported by behavior change theories were found to have good potential to be more effective in improving patient health outcomes. The findings from the survey suggested that there is a need to organize training to improve the utilization of counselling theories and strategies among dietitians when delivering MNT to patients. Dietitians in this study also indicated training and education (90%) to be the main support needed to improve practice. Though guidelines indicate that dietitians should utilize behavior change theories and strategies, these are technical skills that require continuous training. Similar findings were presented by Rapoport, L., and Nicholson Perry, K. (2000), who revealed that dietitians in the study felt that they did not receive adequate training in behavior change skills [30].

The major difference in practices between the public and private sectors were in the arrangement for dietetic consultations. More private dietitians were able to see patients sooner, in less than 2 weeks ($p$-value < 0.001). Unlike public dietitians, private dietitians did not offer group consultation at all ($p$-value < 0.001), and more private dietitians could spend more time during individual consultation of new cases—31–60 min ($p$-value < 0.001). The key factor contributing to the difference in practice is probably due to the different practice setting. The public sector provides the bulk of the healthcare services, with 82% of inpatient care and 35% of ambulatory care. The private sector contributes to about 18% of inpatient care and 62% of ambulatory care [31]. The public sector is heavily subsidized by yearly budget allocations by the government. Patients who seek healthcare services in the public sector pay a minimal fee for the services, and they are mainly low-to-middle-income people. Meanwhile, private health services are provided mainly in urban areas and are offered in either general practitioner clinics or private hospitals. The private sector is funded through out-of-pocket money from patients or private health insurance, who may have higher expectations of the services [32].

*Strengths, Limitations and Recommendations*

This survey is the first step of a plan to develop strategies to support effective dietetic practice for the management of the escalating T2DM. This is an opportunity for the profession to strategize for more comprehensive training to improve the uptake of research in behavioral counselling.

While the survey did not receive the sample size required, the respondent profile reflected broad MDA membership characteristics. A few strategies as described in a review by Nulty (2008) were adopted to boost response rate [33]. Frequent reminders were sent through MDA Facebook and MOH Whatsapp groups. Incentives in the form of e-vouchers were provided to respondents who completed the survey.

There are a few things that future surveys should consider. A question asking dietitians about the common etiology used when formulating nutrition diagnoses should be included. Lewis et al. (2020) discovered that the presence of the etiology–intervention link and the improvement of dietetic care documentation increased the odds of nutrition diagnosis improvement by 52.43% and 37.7%, respectively [3]. Three-quarters of the Malaysian dietitians also reported determining the etiology of a nutrition diagnosis as difficult or very difficult [34]. For better insights regarding training needs, future surveys should also consider questions on the types and content of continuing professional development activities that dietitians would like to learn.

### 5. Conclusions

Training to enhance dietetic practice should revolve around supporting the development of skills in behavioral counseling and strategies, and critical thinking in formulating nutrition diagnosis. To improve the implementation of evidence-based guidelines, the dietitians' practice setting must be considered.

**Author Contributions:** Conceptualization, J.A.J., W.C.S.S. and E.F.M.; methodology, J.A.J. and W.C.S.S.; formal analysis, J.A.J., W.C.S.S. and E.F.M.; writing—original draft preparation, J.A.J., W.C.S.S. and E.F.M.; writing—review and editing, J.A.J., W.C.S.S. and E.F.M.; supervision, W.C.S.S. and E.F.M.; funding acquisition, J.A.J. and W.C.S.S. All authors have read and agreed to the published version of the manuscript.

**Funding:** This research study was funded by International Medical University, grant number PHMS I-2020 (04).

**Institutional Review Board Statement:** The study was conducted in accordance with the Declaration of Helsinki and was approved by International Medical University—Joint Committee on Research and Ethics (protocol code PHMS I-2020 (04) approved on 26 November 2020) for studies involving humans.

**Informed Consent Statement:** Informed consent was obtained from all subjects involved in the study.

**Data Availability Statement:** Data is contained within the article.

**Acknowledgments:** The authors wish to thank Sangeetha Shyam, Fellow at Centre for Translational Research, IMU Institute for Research, Development and Innovation (IRDI), for assisting with the data analysis in the study. The authors expressed tremendous gratitude to all study respondents for their contribution.

**Conflicts of Interest:** The authors declare no conflict of interest.

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
