# Peer review of "Defining the Practice of Dietitians in Malaysia Using the Nutrition Care Process in Patients with Type 2 Diabetes Mellitus"

_2674-0311, doi:10.3390/dietetics1030018_

Round 1
Reviewer 1 Report
The manuscript entitled "Defining the Practice of Dietitians in Malaysia using the Nutrition Care Process in Patients with Type 2 Diabetes Mellitus" is an original survey study to assess the application of the current practices regarding provision of nutrition care in type 2 diabetes mellitus among dietitians in Malaysia. The subject matter undertaken by the authors is important and useful in diabetes practice. Adherence to dietary recommendations is a primary and key therapeutic management in patients with diabetes, especially type 2 diabetes with often associated overweight. A well-controlled course of diabetes is determined precisely by adherence to a dietary regime, appropriate physical activity and, where necessary, pharmacological treatment. To this end, patient training by appropriate diabetes educators, including dietitians, is needed.
According to my assessment, the weaknesses of the work include:
Major issues
1. In the respondent selection procedure, questionnaires were sent to approximately 1,000 dietitians. Only about 17% of the respondents completed the questionnaire, which seems to indicate that the representativeness of the study group was at a low level. This should be included in the limitations of the study. Questions arise: was the percentage of public and private sector respondents who completed questionnaires comparable? Is it possible to provide calculations of sample representativeness in the surveyed group(s) of dietitians at the level of confidence and estimation error?
2. The authors write that "Facial validity was established by piloting a questionnaire with subject matter experts (n = 6)" (line 74). However, in order to validate the questionnaire used, it was necessary to determine its psychometric properties, such as, for example, the Cronbach's alpha internal consistency coefficient.
Author Response
Thank you for the feedback. Please see the attachment.

Reviewer 2 Report
Major comments:
The paper reads well and should be of interest to the dietitians across the world. However, few things need to be considered as follows:
This reviewer suggests that a table to be added to this paper comparing the RDA for Malaysians vs. WHO for macronutrients, vitamins and certain minerals for comparison.
Data analysis needs to be elaborated on. For instance, how were the criteria set and what parameters were grouped together?
Line 62: was the questionnaire validated? Later, the authors indicated that they sent the questionnaire to six individuals. How does this validate the questionnaire? Please explain.
In the discussion or introduction section, the authors need to add a sentence or so to indicated that while CVD is a major concern in this population, there are other conditions that are also affected by diabetes such as poor circulation, neuropathy, DFU, etc.
A copy of the questionnaire or at least the major questions used and analyzed, should be included.
Minor comments:
Any number less than 10 should be spelled out.
Line 30: please change modification to modifications.
Line 32: … assessment steps are shown in Table 3.
Line 35 effectiveness in what? I suggest that sentence to be continued by adding as MNT…
Line 37 if this is a fact, then led should be changed to may lead to…
Line 88: data were entered…
Lines 118-120.. per week respectively at 31-60 minutes for newly referred cases and less 119 than 30 minutes for follow up cases. Suggest moving respectively at the end of the sentence with a comma before respectively.
Lines 155-158: ‘inconsistent carbo- 155 hydrate intake’ (94%), ‘food and nutrition related knowledge deficit’ (85%), ‘intakes of 156 types of carbohydrates inconsistent with needs’… should be changed to , rather than ‘…
Line 160, please add a comma before respectively.
Please type figure 1 (it appears as if it is a picture) also, change the never use to never used to make it consistent.
Author Response

(The authors gave the same response as above.)

Round 2
Reviewer 2 Report
The authors have adequately addressed my comments.